# Spatio-Temporal Analysis of Sawa Lake's Physical Parameters between (1985–2020) and Drought Investigations Using Landsat Imageries

Yousif A. Mousa [1,2,*] , Ali F. Hasan [3] and Petra Helmholz [1]

1 School of Earth and Planetary Sciences, Spatial Sciences, Curtin University, Perth, WA 6845, Australia; petra.helmholz@curtin.edu.au
2 Civil Engineering Department, Al-Muthanna University, Samawah 66001, Al-Muthanna, Iraq
3 Department of Geography, College of Education for Humanity Sciences, Al-Muthanna University, Samawah 66001, Al-Muthanna, Iraq; alifadhil@mu.edu.iq
* Correspondence: yousif.mousa@mu.edu.iq

**Abstract:** Lake Sawa located in Southwest Iraq is a unique natural landscape and without visible inflow and outflow from its surrounding regions. Investigating the environmental and physical dynamics and the hydrological changes in the lake is crucial to understanding the impact of hydrological changes, as well as to inform planning and management in extreme weather events or drought conditions. Lake Sawa is a saltwater lake, covering about 4.9 square kilometers at its largest in the 1980s. In the last decade, the lake has dried out, shrinking to less than 75% of its average size. This contribution focuses on calculating the bank erosion and accretion of Lake Sawa utilizing remote sensing data captured by Landsat platforms (1985–2020). The methodology was validated using higher-resolution Sentinel imagery and field surveys. The outcomes indicated that the area of accretion is significantly higher than erosion, especially of the lake's banks in the far north and the south, in which $1.31 \text{ km}^2$ are lost from its surface area. Further analysis of especially agricultural areas around the lake have been performed to better understand possible reasons causing droughts. Investigations revealed that one possible reason behind droughts is related to the rapid increase in agriculture areas surrounding the lake. It has been found that the agriculture lands have expanded by 475% in 2020 compared to 2010. Linear regression analysis revealed that there is a high correlation (69%) between the expanding of agriculture lands and the drought of Lake Sawa.

**Keywords:** Lake Sawa; spatio-temporal analysis; drought; erosion and accretion; hydrological changes





## 1. Introduction

Understanding the physical characteristics of lake and river drainage systems is important, and it is increasingly of interest in the fields of remote sensing and environmental monitoring. Understanding the physical parameter changes might enable researchers and environmentalists to understand how natural and anthropogenic activities impact those water bodies (e.g., [1]). Lake Sawa, located in the South of Iraq, is of international importance and was a registered wetland in 2014 (no. 2240) by the UNESCO. It is one of the most important permanent water bodies in the area due to its unique, distinct, and natural characteristics, not only nationally but globally [2]. Furthermore, the lake is one of the important Ramsar bodies in Iraq for having several morphological factors.

Firstly, it is a permanent and closed water body with no physical connection, i.e., without any visible water sources in its surrounding area. Instead, groundwater resources (e.g., inner springs or sinkholes) have recently been detected to be the main water supplier [3]. It is claimed that the Dammam aquifers are the main water resource through a system of joints or faults [3].

Lake Sawa is also a point of interest as its water level is significantly higher than its the surrounding area (especially to the southwest) by 2 m higher than the Euphrates River by approximately 5 m [2]. The distance to the Euphrates River (Al-Atshan branch) (North, Northeast, and East of Lake Sawa) is approximately 3.5 km.

Finally, Lake Sawa has an interesting equilibrium system between the gained and lost water capacity and the sodium concentration in the lake [4]. For example, despite the large decrease in water depth (from 5 m to 1.5 m), the salt (Sodium) concentration does not significantly increase. The decrease in the water depth should lead to a rise in the concentration of sodium by at least three times, which is not the case. In contrast, the amounts of sodium increase are neglecting the decrease in the water depth as well as the added inflow of water from the spring (sinkholes). Thus, it is anticipated that the lake has an interesting water recycling system (washing), which could be the second reason for decreasing the water and evaporation [4].

All of those components lead to Lake Sawa characteristics achieving the four natural criteria of the outstanding universal values to be considered World Natural Heritage [2].

The lake's physical parameters (e.g., length, width, shoreline, and area) are the key elements for understanding how they have been affected by global environmental changes over time [1]. Remote Sensing (RS) data and new technologies have proven to be capable and cost-effective tools for studying and monitoring water bodies and their physical dynamics [5]. Moreover, it helps long-term monitoring and planning of the water resources and environmental services [6].

Globally, remote sensing data have often been used for studying and analyzing water bodies, including lakes and rivers. The water bodies were extracted using the Normalized Difference Water Index (NDWI) [7] or the Modified Normalized Difference Water Index within the urban environment (MNDWI) [8]. In Africa, [5] analyzed the surface area variation of Lake Manyara from 2000 to 2011. They extracted the lake body based on calculating the MNDWI from atmospherically corrected data called Moderate Resolution Imaging Spectro-radiometer (MODIS). Other relevant studies considered Tropical Rainfall Measuring Mission (TRMM) and Landsat imagery, e.g., [9]. In addition, [10] extracted inland water surfaces bodies by combining coarse resolution, global water coverage estimates, high-resolution estimates of surface reflection, and topographic information.

In general, the most relevant remote sensing studies focused on observing and monitoring the changes of water levels and/or capacities using the Gravity Recovery and Climate Experiment (GRACE) satellite mission, e.g., [11]; satellite gravimetric and altimetric data, e.g., [12]; or fieldwork, e.g., [4]. While fieldwork is more accurate than space-derived information, it is too expensive and time-consuming for large areas.

Furthermore, several studies have measured the changes in the physical dynamics including areas and shorelines, e.g., [1]. In contrast, some studies analyzed water level and area and volume, e.g., [13]. Moreover, [6] analyzed the physical dimensions of lakes, including their surface area and shoreline, over a 22 year period using multi-mission altimetry and satellite data.

Studies related to Lake Sawa have been conducted and included the chemical properties of the water and/or soil [14–18]. At the same time, very few studies targeted monitoring the water level changes, e.g., [4], and few morphological studies were published in the Arabic language. Even though that remote sensing data have been extensively utilized to study lakes and rivers globally [1,6], it is not the case for Lake Sawa. While [1] studied Victoria Lake's physical dimensions and analyzed its spatio-temporal changes, only four years were used and evaluated. Negligence and insufficient financial support are the main reason for that. This means that a spatio-temporal study of the physical dynamics of a lake over the long term has not been conducted yet.

In recent years, agricultural lands replaced natural pasture lands close to Lake Sawa. Both pasture lands as well as agricultural lands rely mainly on the seasonality of rains. Most of the lands that have been transferred to agriculture are scattered around and do not form one large body of agricultural land. The transformation of pasture to agricultural

lands are coordinated in a form of investment opportunities by the local government and the Investment Authority of the local governorate, and this is accomplished by granting legal investment licenses to investors. Estimates of the Investment Authority made more than 250,000 hectares of agriculture lands available in the Badia regions, i.e., to the Western plateau in which Lake Sawa is located. This agriculture land is used for the production and supply for local and foreign customers. Most of the agricultural areas are planted with strategic crops after preparing the land and digging artesian wells.

The aim of the agricultural investment is to provide agricultural yields, to diversify the income sources of local farmers, and to create job opportunities for the people of the governorate. These have been driving factors in increasing the area of agricultural land in the Badia region after 2010.

The increased agricultural activities make Lake Sawa an interesting lake which needs detailed analysis. As outlined above, remote sensing is an effective tool to do so. To our knowledge, this is the first study on Lake Sawa utilizing remote sensing data for a long-term period study (1985 to 2020). Therefore, the study employs Landsat and Sentinel-2 data to achieve this task. The main contributions of the study are the following:

1. Proposing an accurate method for estimating the length and width of lakes by implementing the so-called Minimum Bounding Rectangle (MBR) method;
2. Determining the lake's physical parameters (surface area and extent) and evaluating its changes since 1985;
3. Identifying the most affected spots (i.e., spots with significant area changes);
4. Estimating the growth in agricultural area around Lake Sawa and assessing their impacts on the drought of the lake.

The remainder of the study is organized as follows. Section 2 introduces the study area, the datasets used, as well as the proposed methodology. The results and analysis are presented in Section 3 before concluding the final outcomes in Section 4.

## 2. Data and Methods

### 2.1. Study Area

Lake Sawa is positioned in Iraq in the western field of the Mesopotamian plain nearby the boundary with Western Sahara. Lake Sawa is located approximately 23 km west of Samawa city southwest of Iraq within 44°59′31″ E–45°01′45″ E and at 31°17′40″ N–31°19′50″ N (Figure 1). The lake consists of limestone rocks and is outlined by gypsum barriers surrounding its border. The chemistry of water is unique, and water salt concentration is extremely high, even higher than the water in the Arab Gulf [19].

### 2.2. Data

In this study, satellite imagery captured by the Landsat and Sentinel-2 were utilized. These images were downloaded from the United States Geological Survey (USGS) website (https://earthexplorer.usgs.gov/ accessed on 20 October 2020) and were already atmospherically corrected (Level-2 products). For the detection of the lake's property, only July images were considered, as the cloud cover ratio is minimal. Additionally, July is summertime in Iraq, and there is no anticipated rain to change the measurements of the water surface area and level in the lake. The images were selected from the Landsat path 168 and row 38. Overall, 31 images from July 1985 to July 2020 were used to analyze spatiotemporal variations in Lake Sawa, and 10 images were used for estimating the agriculture lands (see Table 1).

Landsat images were also utilized for estimating the agricultural areas. However, because the crops are harvested in May and June, the July images are not sufficient to detect agriculture areas. Instead, March and April images are considered. The agricultural activities around Lake Sawa started approximately in 2010. Thus, the time series between 2010 and 2020 were used to conduct and estimate the areas of the agriculture zones. Only March to April 2012 had to be excluded because there are no Landsat data for calculating Lake Sawa's body in the year 2012.

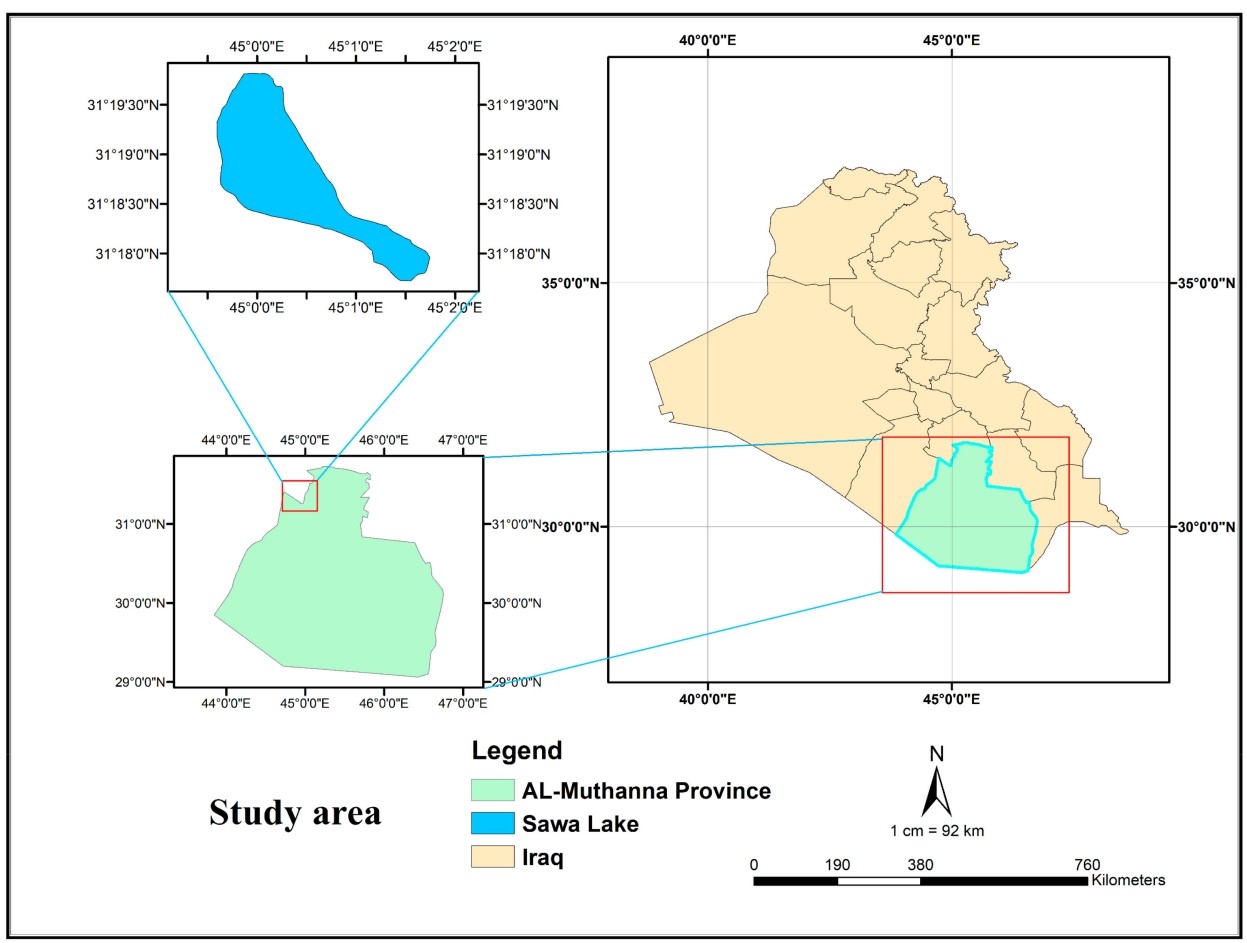

**Figure 1.** The location of the study area.

**Table 1.** Summary of the datasets used for extracting the lake's surface. The data were downloaded from https://earthexplorer.usgs.gov/ (accessed on 20 October 2020).

| | Source | Study Period | Spatial Resolution | Temporal (Days) | Images Used | Usage |
|---|---|---|---|---|---|---|
| Multi-spectral images | Landsat 5 | July (1985–2011) | 30 m | 16 | 21 | Lake's parameters |
| | Landsat 8 | July (2013–2020) | 30 m | 16 | 8 | Lake's parameters |
| | Landsat 5 | March (2010–2011) | 30 m | 16 | 2 | Agricultural Zones |
| | Landsat 8 | March (2010–2020) | 30 m | 16 | 8 | Agricultural Zones |
| | Sentinel-2 | July (2020) | 10 m | 5 | 2 | Lake's parameters |

Satellite images taken with the Sentinel-2 (S2) sensor were used to validate the Landsat results and to prove the suitability of the Landsat-derived characteristics for the monitoring of the spatial coverage within the study area. The delivered Sentinel imagery belongs to Level 1C and has a 10 m Ground Sample Distance (GSD). Sentinel images include 12 spectral bands, but only the visible bands were used in this research. The use of only visible bands was sufficient to digitize Lake Sawa's outline manually to create an accurate reference for the validation purposes. More details of the Landsat and S2 satellite bands utilized are summarized in Table 1.

### 2.3. Rainfall Data

Lake Sawa is characterized by an arid climate with low rainfall ration. Estimating the rainfall ration is very important to understand how climate change impacts the region, which can be another contributing factor for the decline of the lake. The Tropical Rainfall Measuring Mission TRMM from NASA (https://power.larc.nasa.gov/data-access-viewer/ accessed on 16 February 2021) was used to estimate the rainfall in the study area.

### 2.4. Methodology

The applied methodology is presented as in Figure 2. The figure shows three major sections: input data, pre-processing, and the applied methodology. The input data included Landsat (5 and 8) images, Sentinel images, and field work. All images were passed through enhancement and combination processing. Mosaicking procedure was applied only when one image was not sufficient to cover the whole lake surface (e.g., Sentinel images in our case study). Then, Sawa's shorelines and agricultural lands were extracted from Landsat images. Images used for shoreline extraction were captured in July while those used for extracting agricultures were captured in March and April. Alongside these, Sentinel images were used only for digitizing Lake Sawa's shoreline manually for validation purposes. Furthermore, a validation procedure was conducted utilizing the reference as well as the field work measurements. Finally, a regression analysis was performed to find any correlation that caused the drought of Lake Sawa.

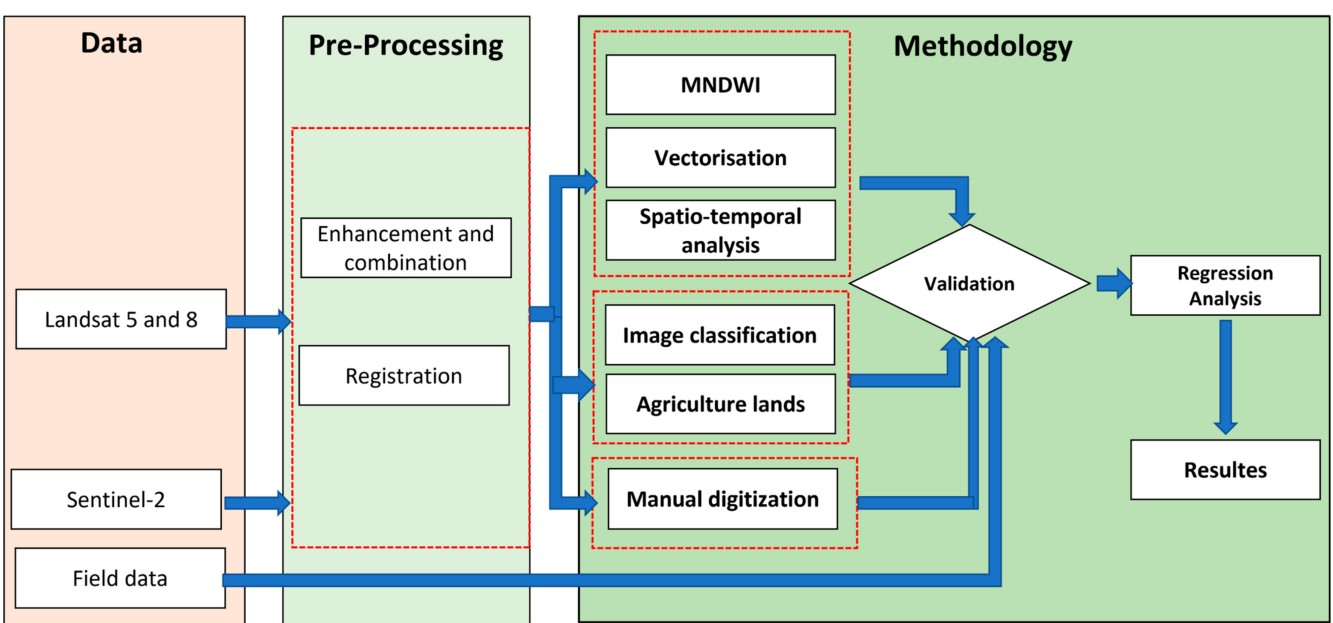

**Figure 2.** Flowchart for describing the applied methodology.

#### 2.4.1. Pre-Processing

As mentioned previously, the used images (level-2) are already atmospherically corrected, except the 1992 ones (level-1). Fortunately, a single satellite image can cover the full extent of the current study site (Sawa). There was only one image (1992) which did not fully covered the lake's area. Alternatively, an image from the Landsat level-1 product was used. This image required an atmospheric correction (e.g., radiometric correction and dark object subtraction), which was performed using [20,21]. Then, registration and clipping procedures were conducted on the Landsat imagery from 1985 until 2020 by clipping all Landsat images to the 2020 image, which was selected as the reference for the clipping mask. The snap raster environment option is performed to align pixels in the clipped patches, considering the adjusted raster image (Landsat 2020) as the reference. Regarding Sentinel images, image combination was performed using semi-automatic clas-

sification Plugin [20,21], which performs atmospheric correction and band combinations. Only three bands were utilized which were sufficient for digitizing Lake Sawa's shoreline manually as a reference for validation purposes.

### 2.4.2. Shoreline Extraction

Rather than using the Normalized Differential Water Index (NDWI), the Modified Normalized Difference Water Index (MNDWI) is applied to the Landsat images to extract the boundary of Lake Sawa. This is because it was proven to be more precise and accurate as the noise of the vegetation and the built-up areas can be mitigated with the assistance of green and medium infrared bands (MIR) [8]. The MNDWI values range was between −1 (vegetation and land surface) and +1 (pure water). MNDWI delivers higher positive values of −1 to +1 (close to one) for water than the near-infrared (NIR) of the NDWI because of the absorption of light [8]. Generally, the MNDWI is calculated using the green and MIR bands represented by bands 2 and 5 (Landsat 5) and bands 3 and 6 (Landsat 8). The required MNDWI equations for Landsat 5 and 7 (Equation (1)) and Landsat 8 (Equation (2)) are presented below. G and the MIR refer to the green and medium infrared bands, respectively:

$$\text{For Landsat 5 (TM) and 7 (ETM+), MNDWI} = \frac{\text{band 2(G)} - \text{band 5(MIR)}}{\text{band 2(G)} + \text{band 5 (MIR)}} \quad (1)$$

$$\text{For Landsat 8 (OLI), MNDWI} = \frac{\text{band 3(G)} - \text{band 6(MIR)}}{\text{band 3(G)} + \text{band 6 (MIR)}} \quad (2)$$

### 2.4.3. Shoreline Shifting

In the context of this paper, erosion and accretion refer to the increases and decreases in the water's surface area, respectively. The lake's depth is approximately graduated from zero (at a shoreline where water meets with land) to about 5 m towards the center. Thus, accretion is always associated with the declining water level of Lake Sawa. Firstly, the spatial union of each consecutive boundary polygons (e.g., 2019 and 2020) is performed to derive a measure for the lake's erosion and accretion. In the newly created polygons (the output of union of each consecutive year), positive values were labeled as accretion, indicating a decrease in the lake body. On the contrary, negative values were used to denote erosion, which indicates an expansion of the lake towards land [22]. If there was no change between the two consecutive boundary polygons, a zero value was given. The technique was repeated for each consecutive datum. Then, the total areas of erosion and accretion are estimated. For measuring the shoreline shifting distance of Lake Sawa's shoreline, 234 transects are manually plotted. These digitized transects were overlayed with the created polygons (e.g., the output of union of each consecutive years) to measure the shifting distance.

### 2.4.4. Agriculture Classification

In general, the groundwater flow of Lake Sawa is most strongly driven by topographic gradients [23]. The local flow of the systems increases with topographic slope (West–East in our case). The agriculture zones are located on the pathways for ground water: the line between regional recharge of water and discharge at the basin of the lake (the topographic slope direction) and in three directions, (Northwest–West–Southwest). Therefore, about 50 km in the three directions was considered to estimate the agricultural areas (see Figure 3).

Landsat images were processed for estimating the agricultural zones using an object-oriented classification method. More specifically, the Example-Based Feature Extraction introduced by [24] was applied. Firstly, a segmentation dividing the image into objects by finding edges and grouping pixels into regions [24] was performed. The critical parameter is selecting the minimum segment size to merge the segments, which was constant for all images. Since the focus was on detecting agricultural lands, only two classes were used to identify the feature classes (e.g., agricultures and all other features including water, bare land, etc.). Second, we chose each feature class and identified the segments that represent

each feature class in Landsat images by visual interpretation. We defined 25 segments each for agricultural zones and non-agricultural zones. Third, we applied the Support Vector Machine (SVM) Classification Method based on spectral, textural, and spatial criteria.

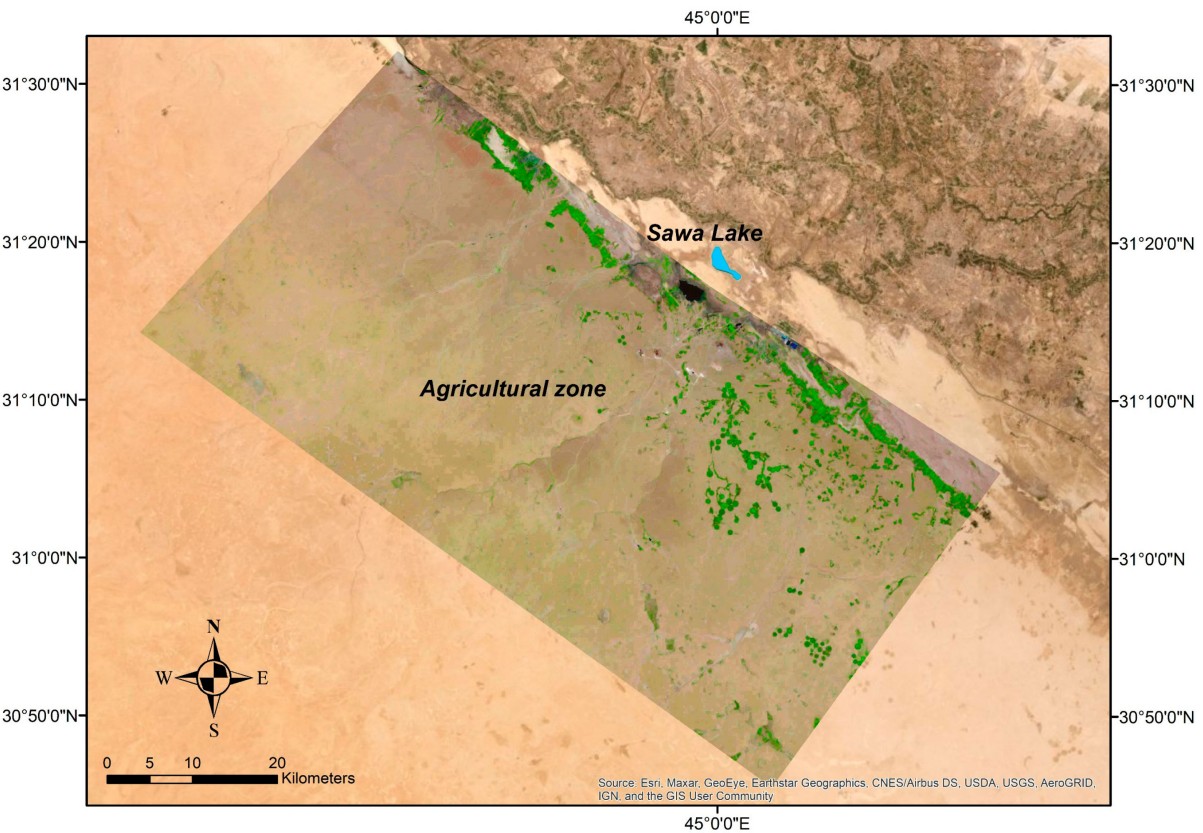

**Figure 3.** The boundary of the selected region that used for estimating the agricultural areas.

*2.5. Accuracy Assessment*

In this section, an accuracy assessment procedure was conducted to validate the applied methodology and its outcomes. This included an assessment for the obtained physical dynamics of Lake Sawa (e.g., area and shorelines) as well as for the obtained agriculture areas.

2.5.1. Shoreline Extraction

The evaluation method applied to validate the outcomes was implemented based on the positional accuracy of the extracted shorelines, e.g., [1,25]. Sentinel-2 images (2020) were utilized for the validation. Sentinel-2 images offer a higher spatial resolution compared to the Landsat used for the spatial-temporal analysis in this work. Sawa is located between two Sentinel-2 tiles (T38RNV and T38RMV), and both are utilized. A mosaicking procedure was applied to create one image covering the whole lake area. Using the created mosaic, Sawa shoreline (polygon) was manually digitized to obtain an accurate reference. It was decided to extract the shoreline manually and not utilizing any automatic approaches to create a reference dataset as accurate as possible for the evaluation. As the outline was digitized manually, the visible bands were sufficient to be used in the study, despite that Sentinel imageries contain 12 spectral bands.

To evaluate the automatically extracted Lake Sawa surface polygon using Landsat, the resulting polygon was compared to the reference polygon, which was created using the Sentinel-2 imagery. For the comparison, the standard metric, e.g., Root Mean Squared Error (*RMSE*), was used. Firstly, all vertices of these two polygons were identified. Since the number of the vertices in both models (extracted and reference) were not equivalent,

calculating the RMSE from vertex (in reference) to vertex (in extracted) did not reflect a valuable evaluation. Therefore, the Vertex to Model evaluation procedure was considered because it is statistically robust and free of subjectivity evaluation [26]. This metric is simply measured the spatial distance from each vertex in a model (polygon) to the closest vertex or edge in the other model (polygon). The distance was calculated utilizing Equation (3):

$$RMSE\,(R,E) = \left( \frac{1}{J+K} \left( \sum_{1}^{J} \left( \min\left( \mathrm{dst}\left( \vec{E,R} \right) \right) \right)^2 + \sum_{1}^{K} \left( \min\left( \mathrm{dst}\left( \vec{R,E} \right) \right) \right)^2 \right) \right)^{0.5}$$

(3)

where, in Equation (3), $J$ and $K$ refer to the number of vertices in the extracted and reference polygons, respectively. Then, based on those distances, the *RMSE* was calculated presenting a single measure for the evaluation of the automatic extracted Landsat polygons.

The maximum error was approximately 91 m (about 3 pixels in the Landsat images). This error occurred in the north side of the lake and effected only a short shoreline section. The estimated RMSE from the reference to the extracted was 25.5 m (Figure 4), which is equivalent of 1 pixel in the Landsat imagery. This result is satisfactory for further analysis. For completeness, we also calculated the RMSE from the extracted polygon to the reference. The result was very similar at 26.8 m. Hence, the applied metric (RMSE) seems to not be affected by the different spatial resolutions and densities.

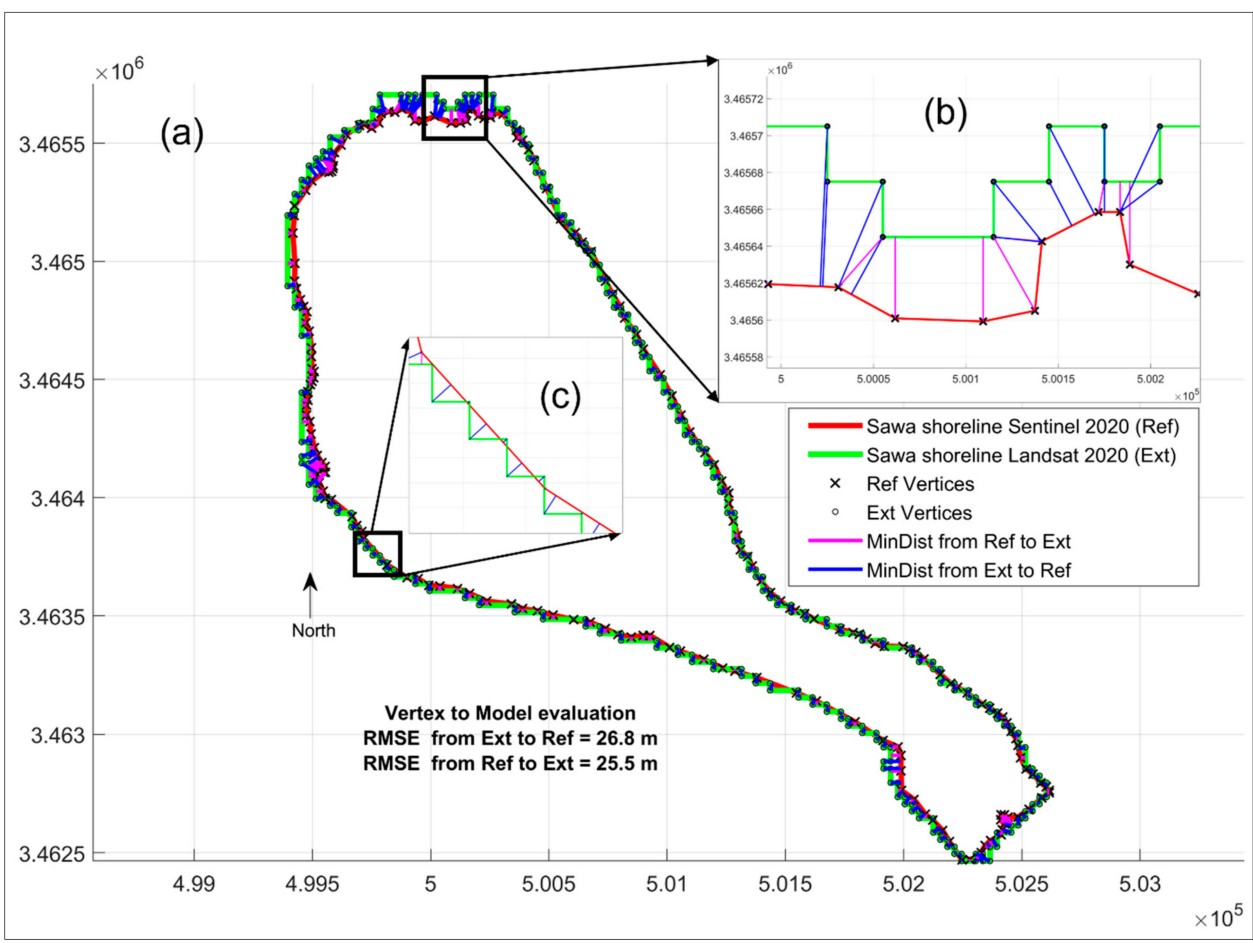

**Figure 4.** Vertex to Model evaluation between the reference Sawa shoreline (red polygon, reference calculated based onSentinel-2 imagery, reference) and the extracted shoreline (green polygon, extracted from Landsat images) (**a**). (**b**) refers to the highest spatial errors with nearly 90 m. While (**c**) illustrates the Zag-Zag form of the extracted shoreline (following pixel edges) compared to the reference (manually digitized) (red line).

The length of the shoreline is approximately 10,309 m using the Sentinel-2 imagery. Again, the observation for Sentinel was used as reference. The automatically detected shoreline length of Landsat is 13,440 m, meaning that an accuracy of about 70% is achieved. The low rate of accuracy can be explained as follows. The extraction of the refence dataset from the Sentinel imagery was performed manually and reflects the shoreline with a greater level of detail compared to the Landsat-extracted shoreline, which follows the pixels (Figure 4b,c). A subpixel extraction of the shoreline using the Landsat imagery was not possible. Hence, the different spatial resolutions of both datasets impact the resulting shoreline length. Therefore, the shoreline length is not used for any further processing.

The impact of the different spatial resolution is neglectable for the area as well as for the dimensions of a Minimum Bounding Rectangle (MBR) [27]. Lake Sawa has irregular shapes, and therefore, it is not clear which direction should be specified to determine the lake's dimensions. Slight changes in the major direction of measurement will lead to changes in the measured length and width. Therefore, the Minimum Bounding Rectangle (MBR) was applied to consider a uniform measuring procedure and avoid subjectivity. The MBR is defined as the minimum box containing the object that needs to be measured in length and width [27]. The MBR allows one to obtain the length and width measurements with no ambiguity. We see the length and the width of the boundary box as an important measure to quantify the change of the lake over time. The MBR is not impacted by the different spatial resolutions of reference dataset (Sentinel) and the Landsat extracted dataset. The width and length of the MBR using the reference polygon (Sentinel) were 1616 m and 4062 m compared to 1672 and 4141 m (Landsat), achieving an accuracy of approximately 97%, which is satisfying for our application.

Not just for the evaluation but also for any further analysis, the lake's area as well as the MBR length and width were used. In terms of the estimated area of Lake Sawa's body, the area calculated from the reference polygon (Sentinel) is 3.46 km$^2$ compared to 3.58 km$^2$ (Landsat), achieving an accuracy of approximately 96%. All used metrics are summaries in Table 2.

**Table 2.** Parameters to quantify the change of Lake Sawa.

| # | Parameter | Description | Unit |
|---|---|---|---|
| 1 | Area | The area covered by water | km$^2$ |
| 2 | Shoreline shift | The shift (maximum) of the shoreline over time. | m |
| 3 | Length | The longest extend of the water surface area in the North–South direction based on finding the Minimum Bounding Rectangle (MBR) | m |
| 4 | Width | Like length but determining the East-West direction of the MBR | m |

### 2.5.2. Agriculture Areas

To validate the results of images classification, we used the agriculture land sample size of 28.1 hectares and the results of the 2020 classification. The 28.1 hectares land sample is located within 31°10′00″N, 45°06′44″E and at 31°08 35″N, 45°07′13″E and is representative for the agriculture areas around the lake. The reference data were collected through fieldwork. The fieldwork took place in March 2020 and aligns with the Landsat images used for the evaluation. The polygons of the different landcover classes were collected using GNSS with an accuracy of 10 m. Metrics utilized for the evaluation are based on the works of [1,25]. The overall accuracy of the classification is 99.5% and kappa is 0.96.

### 3. Results and Analysis

*3.1. Lake Sawa Status*

The physical parameters of Lake Sawa are derived from the remotely sensed data, i.e., from the utilized Landsat and Sentinel-2 imagery. The parameters have been discussed previously and are summarized in Table 2. The physical parameters of Lake Sawa are estimated and presented as in Table 3. The table shows a clear declining pattern in terms of area, length, and width, with averages of approximately 29%, 16%, and 13%, respectively. It has been highlighted that the highest shrinkage happened in the year 2020. While the shoreline length was excluded from further analysis because it was not accurate due to the spatial resolution issue (e.g., Zag-Zag form), which led to enlarging the shoreline length by approximately 30%. In the following sections, we will discuss those physical parameters, except the shoreline length.

**Table 3.** Summary of the calculated physical dynamics of Lake Sawa, including area and MBR length and width.

| Year | Area (Km$^2$) | %Change | MBR Width (m) | %Change | MBR Length (m) | %Change |
|---|---|---|---|---|---|---|
| 1985 | 4.88 | | 1982 | | 4764 | |
| 1987 | 4.86 | −0.4 | 1978 | −0.2 | 4782 | 0.4 |
| 1988 | 4.9 | 0.8 | 1974 | −0.2 | 4764 | −0.4 |
| 1989 | 4.86 | −0.8 | 1972 | −0.1 | 4752 | −0.3 |
| 1990 | 4.88 | 0.4 | 1972 | 0.0 | 4764 | 0.3 |
| 1991 | 4.82 | −1.2 | 1968 | −0.2 | 4716 | −1.0 |
| 1992 | 4.79 | −0.6 | 1972 | 0.2 | 4734 | 0.4 |
| 1994 | 4.76 | −0.6 | 1962 | −0.5 | 4740 | 0.1 |
| 1995 | 4.85 | 1.9 | 1983 | 1.1 | 4740 | 0.0 |
| 1996 | 4.83 | −0.4 | 1978 | −0.3 | 4758 | 0.4 |
| 1997 | 4.75 | −1.7 | 1960 | −0.9 | 4722 | −0.8 |
| 1998 | 4.86 | 2.3 | 1972 | 0.6 | 4746 | 0.5 |
| 1999 | 4.75 | −2.3 | 1956 | −0.8 | 4704 | −0.9 |
| 2000 | 4.62 | −2.7 | 1936 | −1.0 | 4717 | 0.3 |
| 2001 | 4.7 | 1.7 | 1948 | 0.6 | 4741 | 0.5 |
| 2002 | 4.7 | 0.0 | 1948 | 0.0 | 4735 | −0.1 |
| 2005 | 4.72 | 0.4 | 1956 | 0.4 | 4698 | −0.8 |
| 2006 | 4.62 | −2.1 | 1929 | −1.4 | 4699 | 0.0 |
| 2007 | 4.63 | 0.2 | 1941 | 0.6 | 4698 | 0.0 |
| 2010 | 4.43 | −4.3 | 1909 | −1.6 | 4644 | −1.1 |
| 2011 | 4.65 | 5.0 | 1943 | 1.8 | 4698 | 1.2 |
| 2013 | 4.6 | −1.1 | 1943 | 0.0 | 4680 | −0.4 |
| 2014 | 4.53 | −1.5 | 1927 | −0.8 | 4668 | −0.3 |
| 2015 | 4.27 | −5.7 | 1894 | −1.7 | 4548 | −2.6 |
| 2016 | 4.26 | −0.2 | 1885 | −0.5 | 4524 | −0.5 |
| 2017 | 4.19 | −1.6 | 1866 | −1.0 | 4493 | −0.7 |
| 2018 | 4.01 | −4.3 | 1819 | −2.5 | 4397 | −2.1 |
| 2019 | 4.06 | 1.2 | 1828 | 0.5 | 4428 | 0.7 |
| 2020 | 3.58 | −11.8 | 1672 | −8.5 | 4141 | −6.5 |

*3.2. Temporal Analysis*

3.2.1. Surface Area

The table clearly shows the change in the lake's surface over the years. Accordingly, there is a significant decrease in the area during the 36 years between 1985 and 2020. For example, in 1985, the calculated area is 4.88 Mio km$^2$ compared to 3.58 km$^2$. In other words, there has been a decrease of 1.3 km$^2$ in the water surface area, which indicates significant drought by losing 26.6% of the water surface area by 2020 compared to 1985. Our findings align with other sources such as [4,28].

In addition to the values presented in Table 3, the calculated surface area in kilometers squared is presented as a graph in Figure 5, including an added trendline. For 30 years (1985–2015), there was a constant decline in the area by about 12%. However, during this period there were also some years in which the surface area increased, making the overall reduction much less dramatic compared to the period from 2015 to 2018. During these 3 years, there was a dramatic drop by 3%. The decline became even more pronounced from 2019 to 2020, with about 11% of the water surface area being lost. It means that the lake lost nearly as much water surface area in these 2 years (2019 and 2020) compared to the 30-year period from 1985 to 2015, a trend which is unlikely to be only explained by climate change.

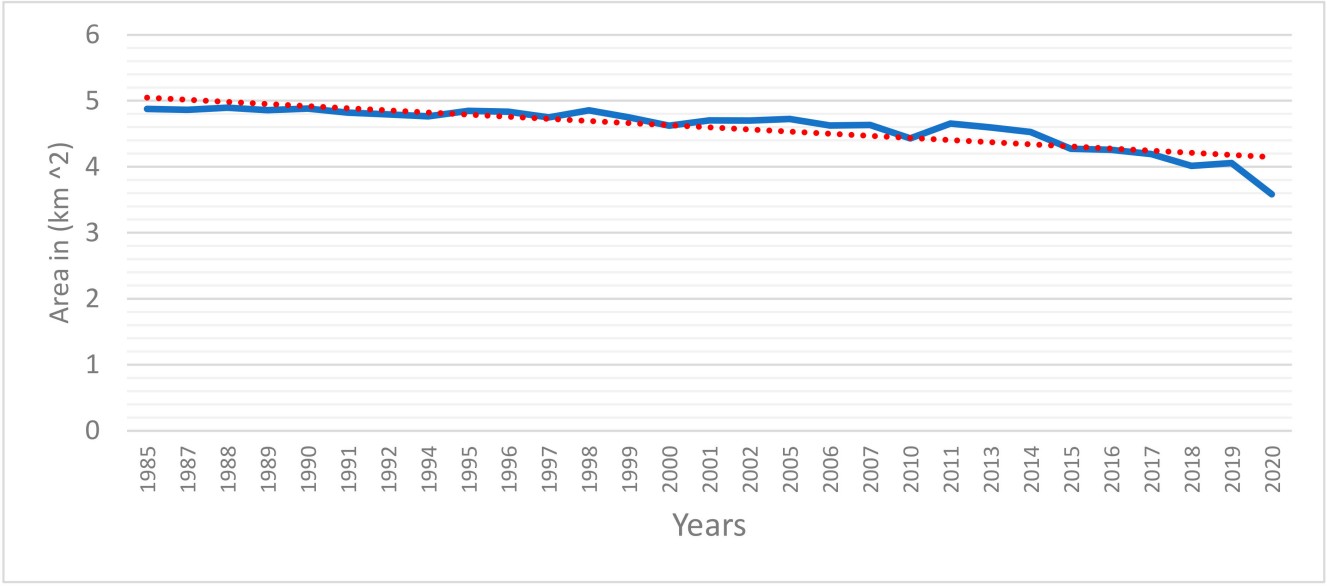

**Figure 5.** The calculated area of Lake Sawa's water surface from 1985 to 2020. The trendline is presented in red dots.

Next, Figure 6 visualizes the area changes of Sawa for each two consecutive years. Erosion is shown in red, and accretion is shown in green. Erosion indicates an increase in the water surface area, associated with an increase in the water level. On the contrary, accretion indicates a decline in the lake's surface area and therefore of its water level. The figure shows the most effected spots, which are located at the north and south of the lake.

Figure 7 shows the erosion and accretion between consecutive years, too, but in histogram form to make it easier to quantify the change. In 2010, there is a high jump in the accretion (e.g., more than 0.2 km$^2$), which is even clearer when considering Figures 6 and 7. The erosion of 2010 is not fully understood due to the lake's complex groundwater supply and will need further investigations. However, in 2011, there was a noticeable increase in erosion reaching 0.22 km$^2$. In 2015, there was again a high jump in accretion by reaching 0.25 km$^2$ associated with almost zero erosion. At the end of 2020, the highest accretion value was unfortunately discovered with more than 0.45 km$^2$. This is a very dangerous indicator for the trend of the droughting of the lake in the next years.

### 3.2.2. Shoreline Shifting

The shoreline shifting between each consecutive years were measured by 234 transects as described in the methodology section. Figure 8 (to the left) depicts the plotted transects labelled by their ID numbers and overlapped over the extracted shorelines 1985 and 2020. On the left, the shifting distances for only five epochs (1985–1987, 1991–1992, 1999–2000, 2010–2011, and 2019–2020) were shown. Positive values refer to accretion while negative values mean erosion. It has been highlighted that the transects numbered (1–20 and 223–234 at the top) indicted the highest shifting of about 350 m between the years 2019 to 2020.

While the transects around 100, which were located at the south of the lake presented about 120 m shifting. Overall, the outcomes completely match with the previous findings that the lake was shrunk noticeably, especially at the north and south spots.

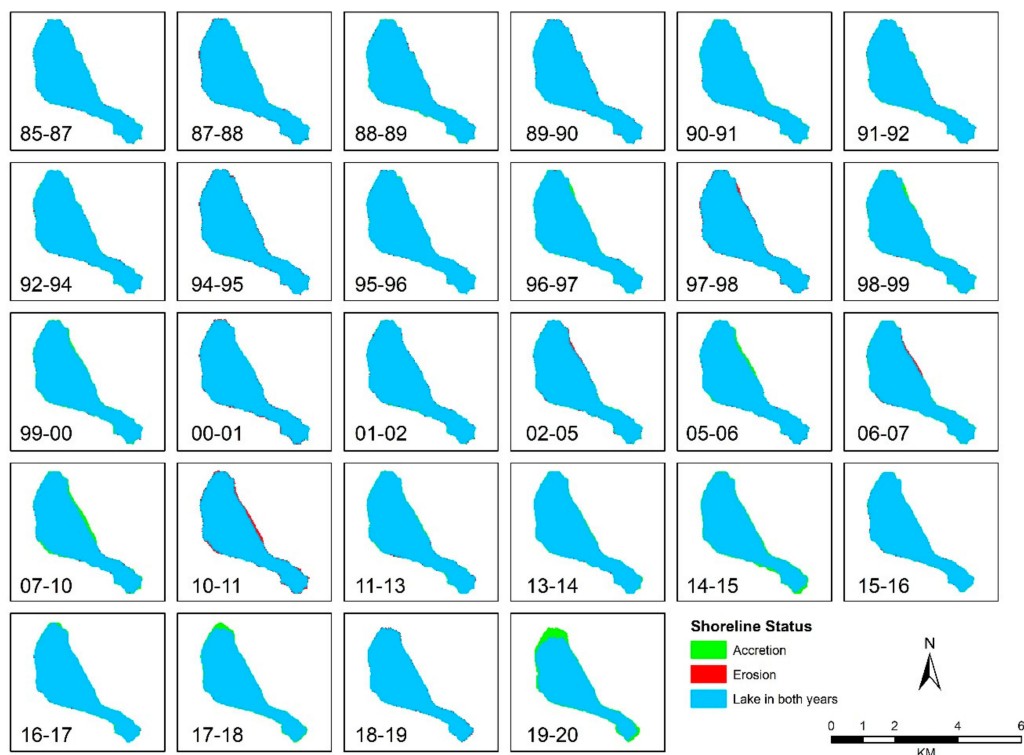

**Figure 6.** Spatial distribution of bank erosion and accretion between for consecutive years between 1985 and 2020.

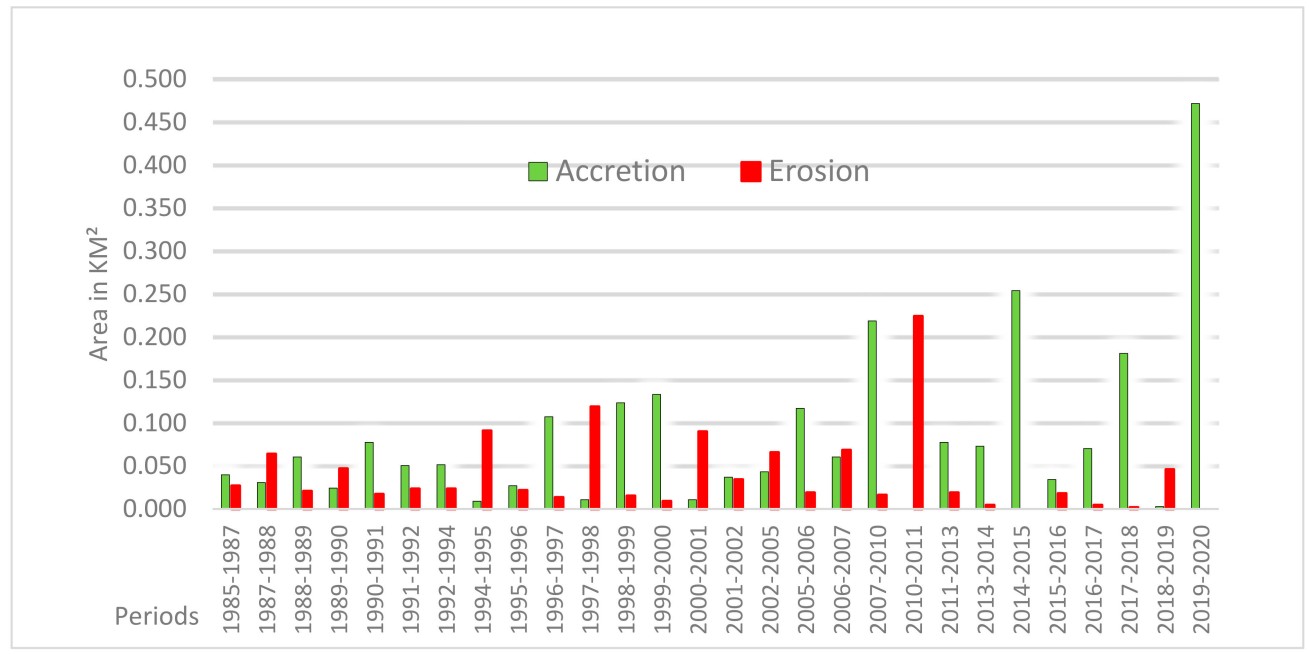

**Figure 7.** The estimated spatial erosion and accretion in Lake Sawa's bank between 1985 and 2020.

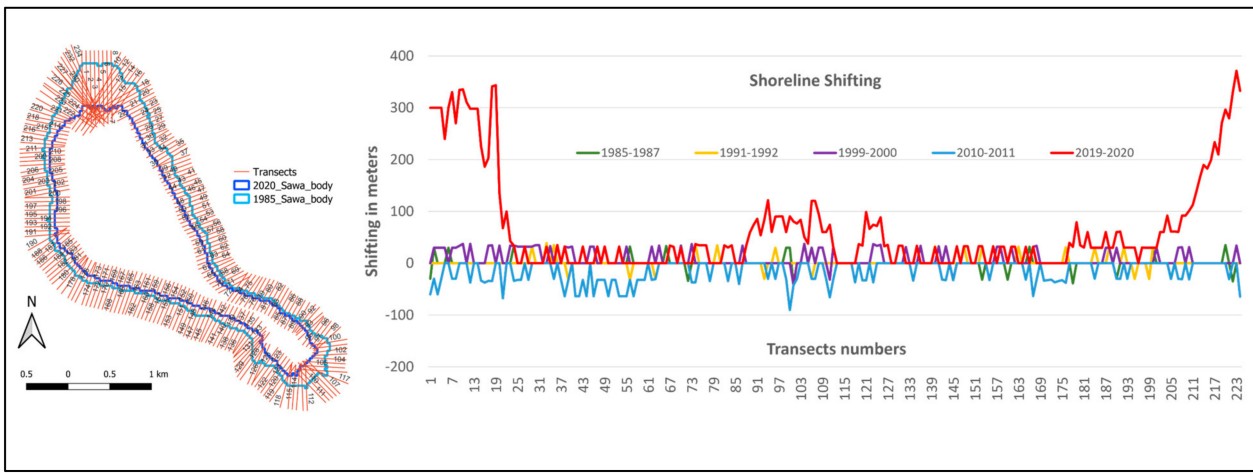

**Figure 8.** The plotted transects and shoreline shifting in meters between each consecutive year. Positive values refer to accretion while negative values mean erosion.

### 3.2.3. MBR Length and Width

The estimated length and width of the MBR are presented in Table 3; a visual representation is shown in Figure 9. There is a significant decline in the length of the lake, with 13% in 2020 compared to 1985. Like the area of the water surface, a constant decline is visible from 1985 to 2015. The estimated width of Lake Sawa is decreasing linear until 2019 with approximately 7.7%. However, there is a noticeable shrinkage appearing in 2020 by 8% compared to 2019. Generally, the decrease in the width is smaller compared to the length because the edges of Lake Sawa's bank in the West and East are very steep (rocky cliff) compared to the gradual deepening of the lake in the North and South.

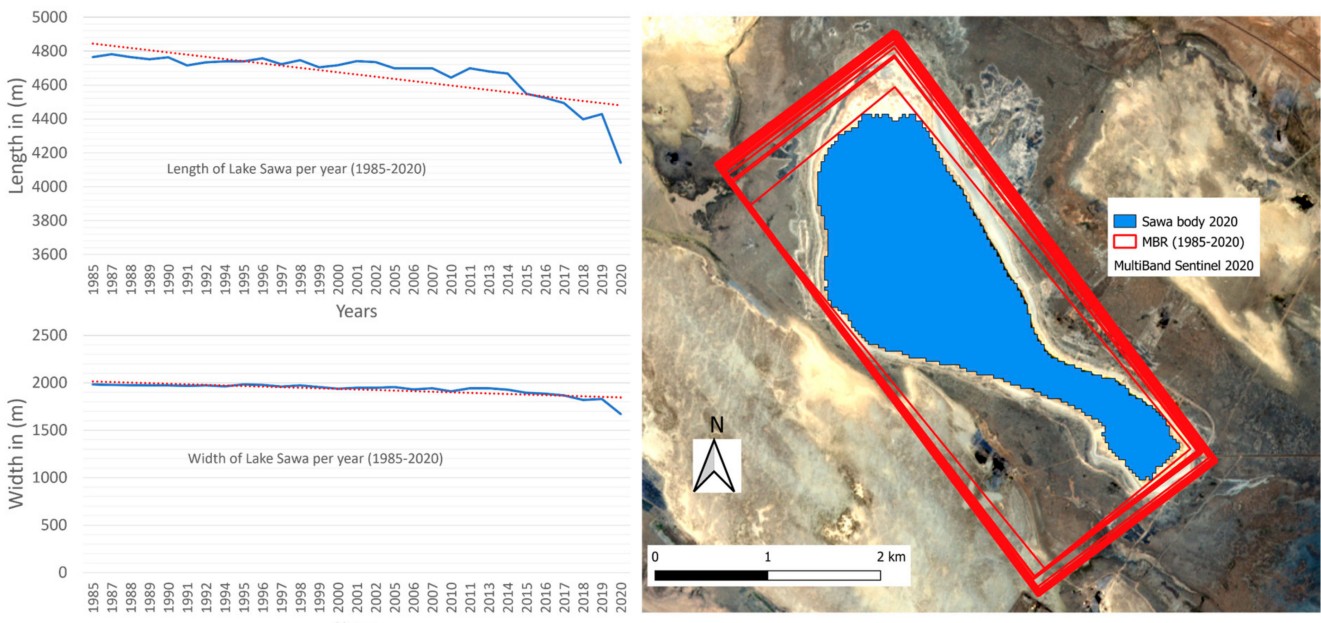

**Figure 9.** (**Left**) Profiles of the estimated length and width of the MBR. (**Right**) Visual representation of the calculated MBR (from 1985–2020).

### 3.3. Agriculture Area

The increases in agricultural areas between 2010 and 2020 are shown in Figure 10; a visual inspection of the figure shows a clear trend of the increase in agricultural activities. While only scattered patches of agricultural land are visible in 2012, a clear increase is

visible with sustainable growth until 2020. The scarcity of water resources is an obstacle for expanding the cultivated areas is known. However, the entire dependency on groundwater might be a possible reason behind the drought of Lake Sawa.

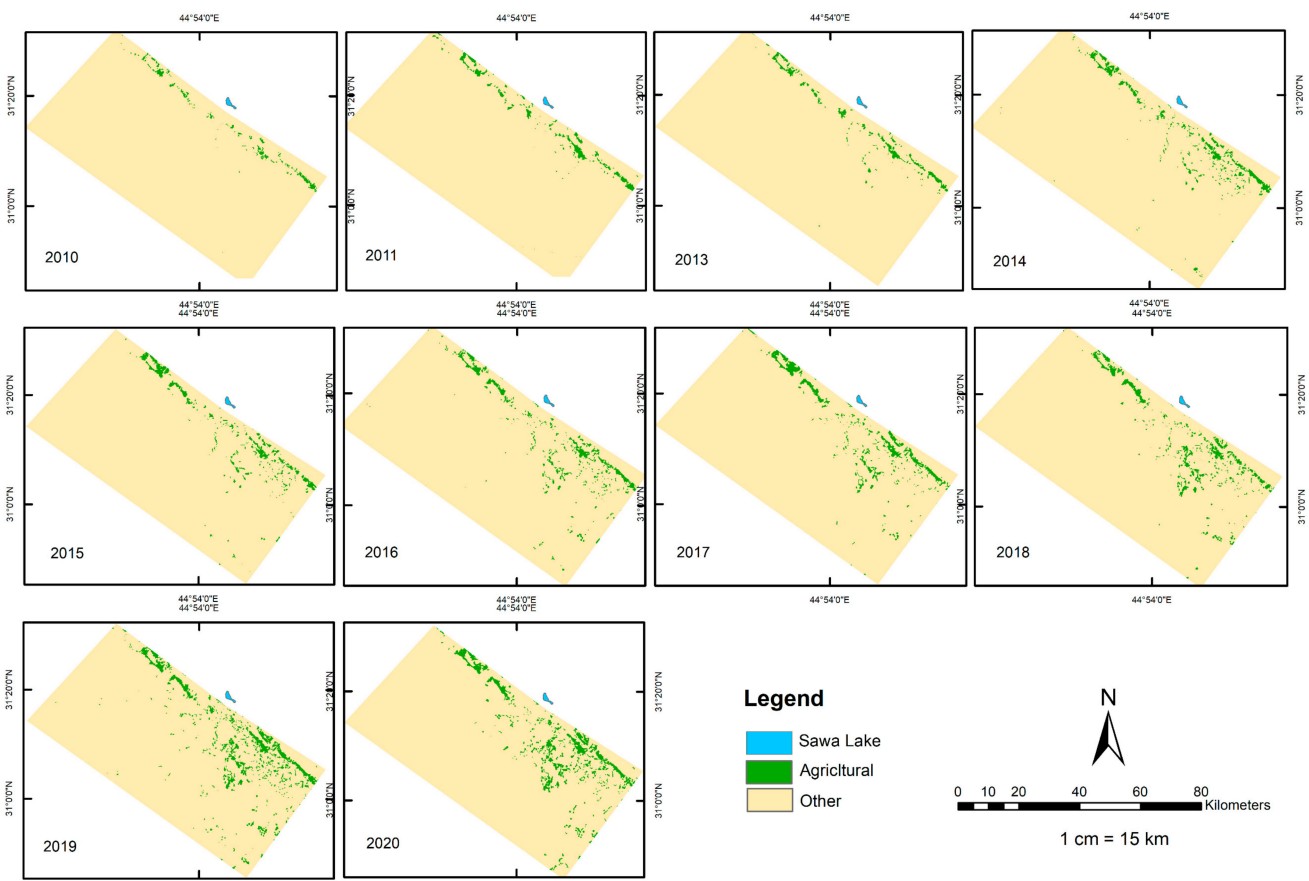

**Figure 10.** The estimated agricultural zones using object-oriented classification of Landsat satellite images (2010–2020) for lands near Lake Sawa. The agricultural lands are represented by green color, while the yellow color refers to the other land uses/land cover classes.

To quantify the growth in agricultural areas, the estimated agricultural lands are presented in Table 4. The results show the extreme expansion of agricultural land since 2010. For instance, the area of agricultural lands increased from 5996 hectares in 2013 to 9381 hectares in 2014, an increase of 56.45%. Similarly, in 2019, the agricultural lands increased by 7000 hectares compared to the previous year (+58.78%). These results indicate a rapid conversion of landscapes into productive farms at the expense of the prevailing ecosystems such as pastures and other plant covers. This is a serious threat of the loss of the prevailing ecological and biological systems, as well as the growing depletion of groundwater, which is the only source of irrigation. Furthermore, the Badia region is characterized by its dryness in most days of the year, with small percentages of the average rainfall, which does not exceed 150 mm in most cases.

This excessive use of groundwater has significantly affected the flows of groundwater that supply Lake Sawa. The data presented in this paper support the theory that the spread of wells near the lake led to an interruption in the underground recharge paths of the spring supplying the lake with water.

**Table 4.** The area of agricultural land and Lake Sawa in hectares, as well as the expanding percentage of agricultural lands compared to the previous year.

| Years | Agriculture Area in Hectares | Lake Sawa Area in Hectares | Expanding of Agricultures Land in % Compared to the Previous Year |
|---|---|---|---|
| 2010 | 3799 | 442.98 | |
| 2011 | 6439 | 465.39 | 69.49 |
| 2013 | 5996 | 459.54 | −6.88 |
| 2014 | 9381 | 452.7 | 56.45 |
| 2015 | 9142 | 427.32 | −2.55 |
| 2016 | 10,482 | 425.7 | 14.66 |
| 2017 | 11,958 | 419.13 | 14.08 |
| 2018 | 11,810 | 401.22 | −1.24 |
| 2019 | 18,752 | 405.54 | 58.78 |
| 2020 | 18,059 | 358.38 | −3.70 |

### 3.4. Rainfall Data Analysis

Next, the correlation with the rainfall data is investigated. The data were acquired in monthly intervals for the period from 1985 to 2020 (Figure 11). The mean annual rainfall is 150 mm per year (1985–2020). The rain fluctuates up and down with major rainfall events, leading to above-average rainfall totals in the last three years. The above-average rainfall events contrast with the drastic decline in the lake's surface areas during the same time period. Therefore, the lack of rainfall is clearly not the reason for the drought of Lake Sawa.

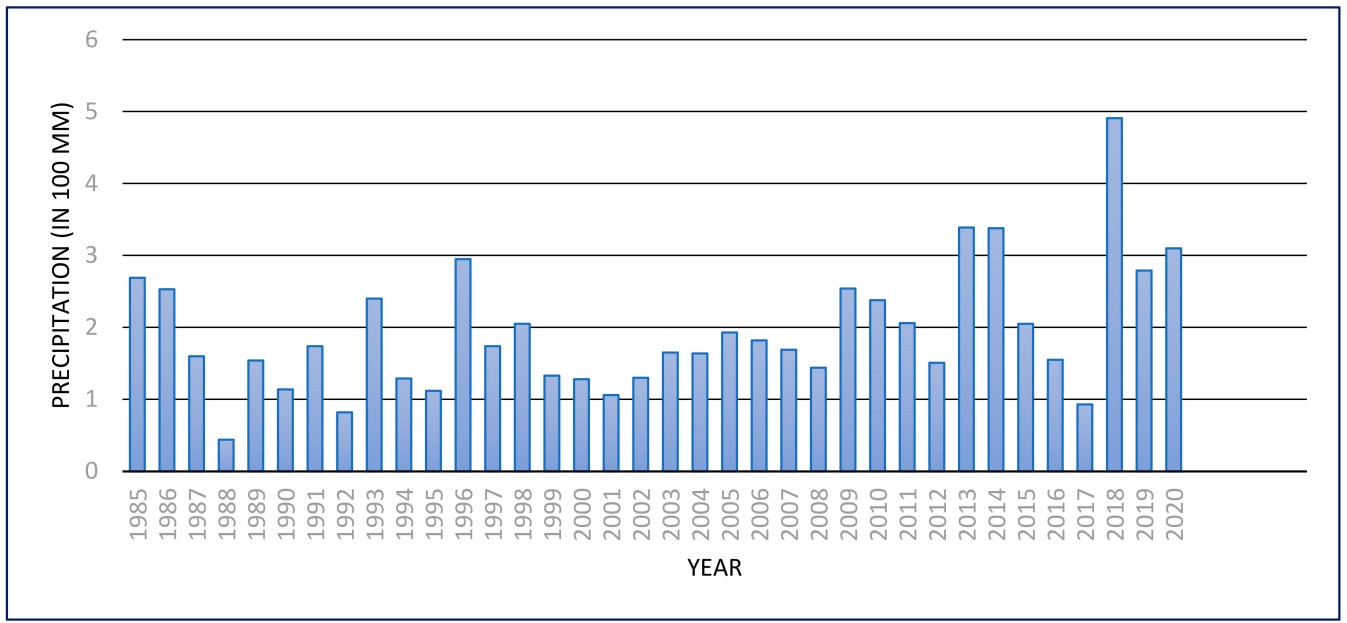

**Figure 11.** The precipitation in 100 mm of study area (Sawa). Data source is (https://power.larc.nasa.gov/data-access-viewer/ accessed on 16 February 2021).

### 3.5. Linear Regression Analysis

Many theories have been raised by experts and participants in monitoring and interpreting the causes of the shrinkage of Lake Sawa. Some claimed that shifting tectonic plates from earthquakes also demolished the main tributary feeding the lake, or the weather could be a contributing factor due to low rainfall and increasing evaporation. In this contribution,

the relationship between Lake Sawa's drought and agricultural activity augmentation has been identified as a reason and will be further underlined by a linear regression analysis.

Figure 12 shows a plot of the total coverage of agriculture areas and the lake's total surface. The Figure 12 presents a rapid expanding in agricultural lands (green line) alongside a dramatic decline in Sawa' size (red line). Additionally, the Badia region close to Sawa contains number of industrial zones for stone and sand quarries and the production of table salt, which also requires large quantities of groundwater, but which have not been quantified in this paper. Estimating the amount of water that is withdrawn from wells for industrial purposes is very difficult due to the lack of accurate information.

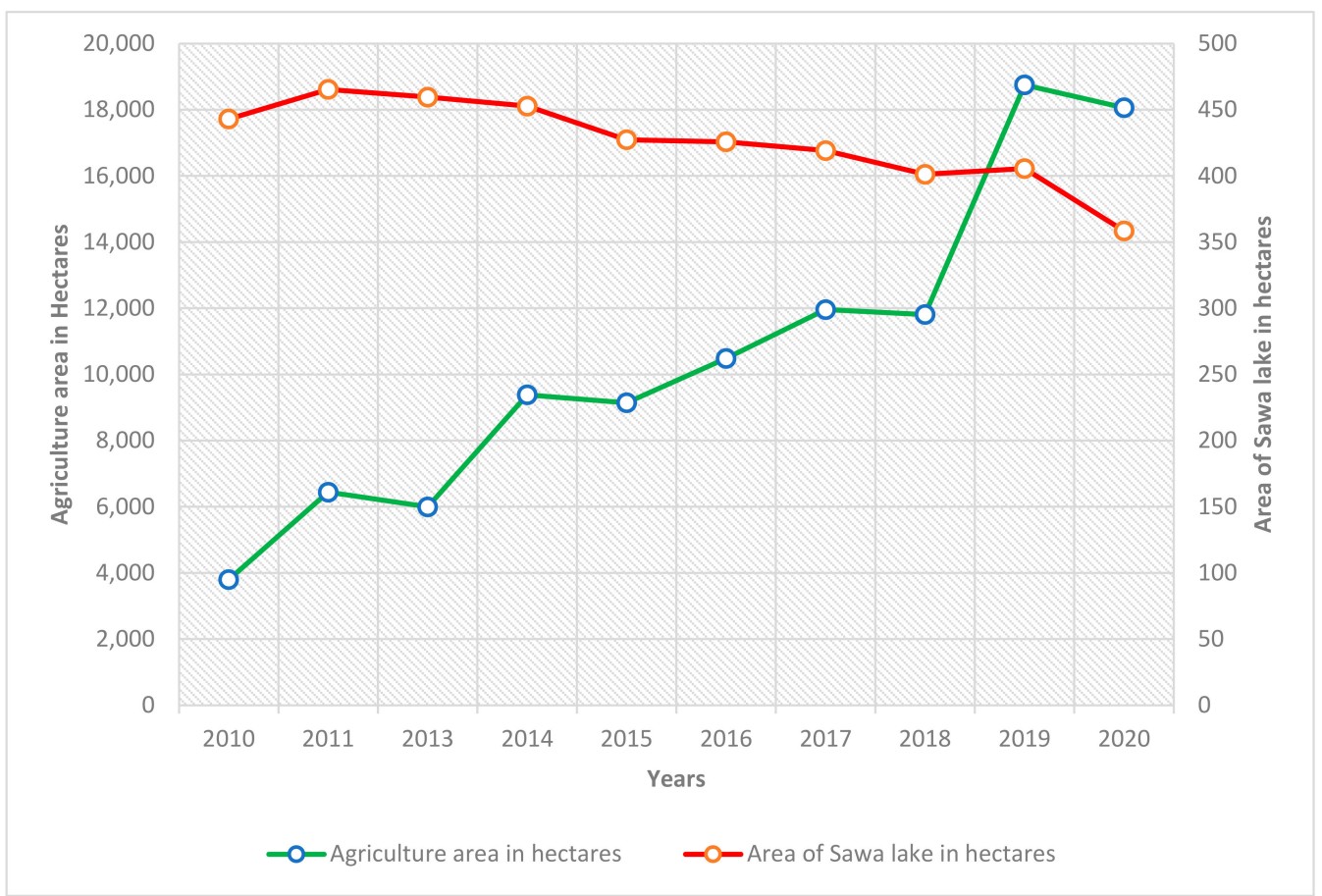

**Figure 12.** The changes in the Sawa basin area and agriculture areas for the 2010–2020 period.

A linear regression analysis has been conducted to find the relationship between the changes in the Lake Sawa area and the agricultural areas; the result is presented in Figure 13. The analysis shows a strong correlation of approximately 70% ($R^2$ = 0.6965), indicating that the increase in agriculture areas leads to a drying up of Lake Sawa. This is indeed caused by the increasing rush to consume groundwater resources. Therefore, this rapid increase in agricultural lands could be one factor causing the quick disappearance of Lake Sawa. Consequently, the lake's disappearance might be looming after this dangerous shrinking alarm, and the important question is whether it will disappear or not.

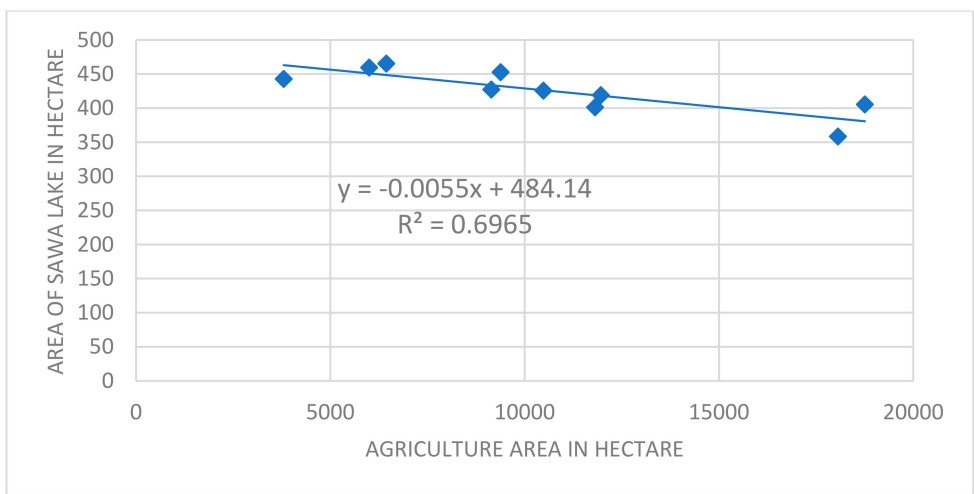

**Figure 13.** Linear regression relationship between the Sawa basin area and agriculture areas for period (2010–2020).

## 4. Conclusions

A long-term spatio-temporal analysis has been conducted for investigating the major reasons that caused the dangerous shrinking of Lake Sawa. On one hand, the physical parameters, e.g., area, shorelines, length, and width have estimated for 29 epochs from 1985 to 2020 using Landsat imagery. Moreover, the area of erosion and accretion have been calculated by carrying the spatial union of each two consecutive boundary polygons utilizing 234 transects or profiles that are manually digitized. While there was a kind of balance between erosion and accretion until 2010, there was a jump in 2015 in accretion reaching to 0.25 km$^2$ (5% compared to 1985) associated with almost zero erosion. The highest accretion value was found in 2020 with more than 0.45 km$^2$ area Sawa (10% compared to 1985). The northern and southern parts of the lake were the most affected spots by the drought because these sections are shallow. Alongside, a new methodology for estimating accurate length and width of the lake has been applied by implementing the Minimum Bounding Rectangle approach. A significant decline is shown in the length of the lake, by approximately 13% in 2020 compared to 1985. All these outcomes indicate that a serious problem is facing the existence of Lake Sawa.

On the other hand, the agricultural lands around Sawa in the western and southern directions have been estimated from 2010 to 2020 using the object-oriented classification method. It has been noticed that the area of the agriculture lands has been extremely expanded annually. For instance, the area of the agricultural lands jumped from 5996 hectares in 2013 to 9381 hectares in 2014 (56.45%). Similarly, in 2019, the agricultural lands have increased by 7000 hectares compared to the previous year (58.78%). These results indicate a rapid conversion of landscapes into productive farms at the expense of the prevailing ecosystems such as pastures and other plant covers. This is a serious threat of the loss of the prevailing ecological and biological systems, as well as the growing depletion of groundwater which is the only source of irrigation.

For the evaluation purposes, Sentinel-2 images have been used as well as fieldwork investigations. As for a ground truth (reference) of Sawa, manual digitization has been considered to achieve valuable validation using Sentinel-2 (2020). To assess the spatial displacement of the automatically extracted Sawa's polygon compared to the reference polygon, the standard topographic measure, e.g., Root Mean Squared Error (RMSE), has been utilized following the Vertex to Model (V2M) evaluation technique. Accordingly, the estimated RMSE from the extracted polygon to the reference and vice versa were 26.8 m and 25.5 m, respectively, and the maximum error between the two models was approximately 91 m (about 3 pixels). However, the absolute RMSE (according to Equation

(3)) was approximately 26 m, which means that the accuracy is within the accepted range of 1 pixel (30 m length and width).

In this study, some reasons that caused the shrinking of Lake Sawa have been investigated. Firstly, it has been found that the problem is not related to changes in the climate, i.e., in rainfall. For instance, it has been found that the mean annual rainfall has fluctuated up and down with a balanced trend. Thus, rainfall factor does not have a clear impact on the drought of Sawa. Therefore, the effects of the rapid expansion of the agriculture lands on Sawa drought has been investigated. To perform this task, a linear regression analysis has been conducted. Our findings refer to a strong correlation of approximately 70% between of the two variables (increased agriculture and drought of the lake), which indicates a possible effect of expanding agriculture lands on the drying up of Lake Sawa. Due to the quantitative results presented in this research, it is unlikely that it is only a story of climate change. The data presented in this paper support the theory that the spread of wells near the lake led to an interruption in the underground recharge paths of the spring supplying the lake with water. Further contributing factors which were outside of the scope of the paper include the overuse of water for stone quarries and the cement and salt industries that have spread in the region over the past years, too. Environmentally, rethinking the use of water resources in this desert region is important to save this water body rich in unique ecological diversity.

**Author Contributions:** Conceptualization, Y.A.M.; methodology, Y.A.M. and A.F.H.; software, Y.A.M. and A.F.H.; validation, Y.A.M. and A.F.H.; formal analysis, Y.A.M. and A.F.H.; investigation, P.H.; resources, Y.A.M. and A.F.H.; data curation, Y.A.M.; writing—original draft preparation, Y.A.M.; writing—review and editing, Y.A.M. and P.H.; visualization, Y.A.M. and P.H.; supervision, P.H.; project administration, Y.A.M.; funding acquisition, P.H. All authors have read and agreed to the published version of the manuscript.

**Funding:** This research received no external funding.

**Acknowledgments:** This study is supported by Curtin University and Al-Muthanna University (Badia Research Centre and Lake Sawa). The authors would also like to acknowledge the editors and reviewers for their valuable comments.

**Conflicts of Interest:** The authors declare no conflict of interest.

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
