# Peer review of "Spatio-Temporal Analysis of Sawa Lake’s Physical Parameters between (1985–2020) and Drought Investigations Using Landsat Imageries"

_remotesensing, doi:10.3390/rs14081831_

Round 1
Reviewer 1 Report
Please find the comments in the enclosed PDF-document.

Author Response
Response to Reviewer 1 Comments
ABSTRACT – The abstract is quite good
INTRODUCTION – Very good written, cited and clear to follow. Aims of the study are nicely communicated.
Thanks a lot for the positive feedback.
DATASETS AND METHODOLOGY
Method section can be improved by describing the methods a bit more in detail, e.g. it is not absolutely clear why Landsat AND Sentinel-2 are used side-by-side. You have not described your conducted fieldwork
The dataset section as well as the methodology section has been improved. For instance, the data section was improved and details about the image data was added. Furthermore, information of the rainfall data was added, too. The methodology section was improved by adding a workflow figure, providing more details to the pre-processing section and sustainable increasing the assessment section. More details are provided below when addressing the specific comments.
RESULTS AND DISCUSSION
At some/many points, sections rather belong to the methodology section. Please check. Some results are quite isolated and not described well. There is only a reference to a figure/table but without description of what the reader can see.
We have considered this feedback which is also further outlined in the specific comments provided by the reviewer. The methodology section was improved sustainable. Results have been reviewed and further descriptions have been provided. All figures and tables have been analyzed and additional text has been added.
CONCLUSION
The conclusion is somewhat very subjective. The authors should avoid statements as that the changes of lake Sawa are not related to climate change. There is no proof for it.
The conclusion has been revisited and we agree that some of the statements were to bold. The section has been changed.
GENERAL
- Formatting: formatting especially of citations needs to be improved
- Structure: no need for improvement
SPECIFIC COMMENTS:
Point 1: Why have the authors chosen to use only images of July? Is there no interannual change in the lake dynamics? Could you proof that no fluctuation exists within a year?
Response 1: Thanks for this question. Of course, there is interannual change in the lake, but the images of July were chosen because the cloud ratio is smallest during the whole year. Additionally, July is summertime in Iraq and there is no anticipated rain to change the measurements of the water surface area and level in the lake. [This information has been added to section 2.2.]
Point 2: Why have you only used visible bands of Sentinel-2?
Response 2: The main reason for using Sentinel-2 images is for validation purposes. For example, the physical parameters of the Lake (e.g., area of water surface, shoreline, length and width) were obtained by utilizing the Landsat imagery that have 30m * 30m resolution. Therefore, we need a ground truth to validate the result e.g., manually digitized of Sawa lake border from higher resolution (10m * 10m). The visible bands were sufficient to digitize the border manually and compared it to the one automatically obtained border which obtained from Landsat images. [A reason has been added to section 2.2.]
Point 3: Please add the number of used images per Sensor in Table 1
Response 3: The number of images used has been added to table 1 (see below).
|
|
Source |
Study period |
Spatial Resolution |
Temporal (days) |
Images used |
Usage |
|
Multi-spectral images |
Landsat 5 |
Jul (1985-2011) |
30 m |
16 |
21 |
Lake's parameters |
|
Landsat 8 |
Jul (2013-2020) |
30 m |
16 |
8 |
Lake's parameters |
|
|
Landsat 5 |
March (2010-2011) |
30 m |
16 |
2 |
Agricultural Zones |
|
|
Landsat 8 |
March (2010-2020) |
30 m |
16 |
8 |
Agricultural Zones
|
|
|
|
Sentinel-2 |
Jul (2020) |
10 m |
5 |
2 |
Lake's parameters |
Point 5: Section 2.3.3: Can you explain more in detail why you have introduced the transects?
Response 5: Thanks for this question. The transects are perpendicular to the lake’s surface and have been introduced to measure the shoreline shifting (Accretion and Erosion) between each consecutive year perpendicular to the shoreline. We believe that the direction perpendicular to the shoreline expresses best the shift of the shoreline due to the complex shape of the shoreline. However, we have added modified figure shown specific years. Also, the explanation why transects have been used, and how the lake’s erosion and accretion is determined using the transects. (section 3.3.1 and Figure 8).
Point 6: Line 202/203: Can you explain how you evaluate the agricultural land by average sample size?
Response 6: Thanks for the comment. We added a new section outlining exactly how the evaluation was performed. “To validate the results of images classification, we used the agriculture land sample size of 28.1 hectares and the results of the 2020 classification. The 28.1 hectares land sample is located within 31° 10’ 00’’ N, 45° 06’ 44’’ E and at 31° 08’ 35’’ N, 45° 07’ 13’’ E and is representative for the agriculture areas around the lake. The reference data was collected through fieldwork. The fieldwork took place in March 2020 and aligns with the Landsat images used for the evaluation. The polygons of the different landcover classes were collected using GNSS with an accuracy of 10 m. Metrics utilized for the evaluation are based on the works of [25] and [1] The overall accuracy of the classification is 99.5% and kappa is 0.96. “
Point 7: Line 220: reference to Table 2 is missing.
Response 7: Thanks for this out. The reference has been added.
Point 8: Section 3: Name it “Results & Discussion” instead of “Results & Analysis”
Response 8: Thanks for the comment. We believe that both titles, i.e. “Results & Discussion” and “Results & Analysis” reflect the content of the chapter. As none of the other reviews have comment on the chapter title, we have decided to keep the name.
Point 9: Table 3: add percentage change or/and up- and down-arrows to the table
Response 9: Thanks for the feedback. We have considered the feedback and have updated the table.
Point 10: Table 3: consider moving Table 3 to your supplementary material
Response 10: We have considered the feedback and have decided to keep the table in the result section. We see it as important to present the data to the readers in the main section. We see our decision supported by the other two reviewers who have not point out this issue. However, we do agree that the importance of presenting the data has not been clear. Therefore, we have added to this section and have discussed the table in the text in more detail.
Point 11: Section 3.1.: Only shows the tables/figures but do not properly explain what can be seen. Please change.
Response 11: Thanks for the feedback and we agree that the discussion of the tables/figure have been too short. We have added rational to the measures shown in Table 2, and we have added a discussion of Table 3.
Point 12: Section 3.2.: Belongs rather to the method section
Response 12: Thanks for this note, the section has been moved to the method section
Point 13: Figure 4: In my opinion it is rather a question of different formats (raster vs. vector) and the spatial resolution of the satellite imagery. How can you justify it?
Response 13: In the original manuscript we didn’t discuss the issue of the different spatial resoultions. Please note that the Landsat derived dataset is also a vector file but with a reduce spatial resolution compared to the Sentinel dataset. We have added a discussion related to the different spatial resolutions to the manuscript. Furthermore, after reviewing the data, we have concluded to exclude the metrics of the shoreline length as it is heavily impacted by the different spatial resolutions of the datasets. In contrast, the RMSE, the area of the lake and also the MBR are not impacted by the different spatial resolutions and have been used for the temporal analysis.
Point 14: Section 3.3.1.: Can you present a trend here?
Response 14: We are unsure that the review means. A trendline was shown and the trend is discussed.
Point 15: Line 288/301: do you mean “trendline” instead of “tradeline”?
Response 15: Thanks for this note. The word has been corrected.
Point 16: Line 297: you mentioned earlier that the year 2020 is your baseline year. Why do you refer to 1985 here?
Response 16: We apologies the misunderstanding. 2020 is not the baseline year. It is the year which has been used to validate the methodology of area and MBR extraction. We had to select 2020 as the validation required field data which was only available for 2020. The base year for the analysis is 1985 as this is the first year which is included in the analysis.
Point 17: Line 304: avoid “last year”
Response 17: Changed to “the year”
Point 18: Line 325: avoid such a repetition
Response 18: We have removed this repeat and have also review the start of the other sections for reviews. When ever possible, those repeats have been removed.
Point 19: Section 3.5: Too much introduction for a section within “Results and Discussion”. Move the introduction to Section 1.
Response 19: We have moved the information about the agricultural development of the area (overall two paragraphs) to the background section as it forms part of the motivation of this study.
Point 20: Figure 20: The maps are too small to read.
Response 20: Figure 11 has been replaced by (Figure 10 in the revised manuscript) with a higher resolution figure, and an extra and enlarged figure of the comparison of 1985 to 2020 has been added
Point 21: Table 4: Move to supplementary material
Response 21: Similar to Table 3 we believe it is important to show the table in the main part of the paper as it quantifies the changes over time. To make the importance of the Table clearer to the reader, extra analysis and discussion has been added to the manuscript.
Point 22: Section 3.6.: Why have you not mentioned these data in Section 2.2.?
Response 22: The rainfall data have been added to section 2 (data)
Point 23: Line 422: please consider writing “ONE factor” instead of “the […] factor”
Response 23: The feedback has been considered and the wording has been changed.
Point 24: Line 473: tough statement. Consider avoiding.
Response 24: Reviewing the manuscript we agree that some of the statements have been to bold. We have revisited the last section of the paper and have change it accordantly.
Point 25: Line 476: “think” instead of “believe”
Response 25: The sentence has been modified.
Reviewer 2 Report
This manuscript conducted a spatio-temporal analysis of the environmental, physical dynamics, and hydrological changes in Sawa Lake in Iraq. The topic is significant and methodology seems to be reasonable. However more throughout analysis should be done when demonstrate the results. More citiations are needed and the structure of the manuscript could be improved. Grammar needs to be fixed as well (some grammar errors were pointed out in my comment below but the authors should check all the pages). Overall the manuscript needs major revisions before it could be considered for publication at the journal of Remote Sensing. Please refer to my comments below for more details.
Lines 1-2: It is unclear what factors about the lake that authors intended to study from the title. The tile states “spatial-temporal analysis of Sawa Lake”, so what about the lake the study wanted to emphasize, e.g., physical conditions, agricultural land change?
The word “this” was used too frequently in the manuscript. Most of them would be better replaced by the word “the”, e.g., in Page 1, Line 34 “in this area” is better to be written as “in the area”. In Line 233, “For this validation purpose…” is better to be written as “For the validation purpose…”. There are many other “this” that the authors should check and replace as well.
Lines 44-47: Any citation for the distance to the Euphrates River? Any citation for statement “Lake Sawa has an interesting equilibrium system…” What do you mean by saying the lake has an interesting equilibrium system? Please explain what is interesting.
Line 50: change “…proved that Lake Sawa has…” to “…proved that Lake Sawa had…”
Line 64: change “…The water bodies are…” to “…The water bodies were…”
Line 86: change “… and includes…” to “…and included…”
Line 95: It would be better to change to “…spatio-temporal changes, for which only four years were included in the study.”
Lines 123-148: It is fine to briefly mention that satellite imagery would be used in the study. However the details would be better placed as part of the methodology. I suggest moving the contents in those lines (123-148) to the section of Methodology/Pre-processing.
The authors need to check the sentences in the methodology section. Past tense instead of present tense should be used to describe the steps that the authors conducted for the project.
Line 152: change “…the quality of this image (1992)…” to “…the quality of the image (1992, level-1)”.
Lines 153-154: If I understood correctly, there were one level-1 1992 image and one level-2 1992 image used in the study. If it was the case, however, the methodology should clearly label each 1992 image, for example, changing “…mosaicking was applied to the 1992 image, the only time when one image was not sufficient to cover…” to “…mosaicking was applied to the another 1992 image (level-2), the one time when the level-2 1992 image was not sufficient to cover…”
Line 181: change “…indicating decreasing…” to “…indicating a decrease in…”
Line 230: change “…This is including…” to “…This includes…”
Lines 233-255: It is better to place the descriptions for procedure in the Methodology section, and mainly include the results in the Results section.
Line 278: Do you have documents to support your assessment (Signiant drought)?
Line 286: so what other factors could contribute to the trend (a decrease in lake size) besides climate change?
Line 294: change “…less than the automatically…” to “…less than the one automatically…”
Lines 296-299: What were the possible reasons for the shoreline shrinkage?
Lines 330-331: Why “there was a noticeable increase in erosion”?
Lines 385-392: Should these sentences be placed in the Methodology section?
Lines 467-478: please consider changing “…rainfall is fluctuating up…” to “…rainfall has fluctuated up…with a balanced trend since year XXXX…”
Reviewer 3 Report
- The title of this paper is not comprehensive, e.g. Spatio-temporal analysis of what?
- The abstract is not comprehensive, neither the methods are comprehensive nor the results are so appealing.
- Line 21-22: What does the author refer to as the reason for the increase in the agricultural area surrounding the lake?
- There are vague terminologies throughout the manuscript such as ... Thirdly, lake SAWA has an interesting equilibrium system... Line 46
- The references given in the manuscript are not synchronized, for example, line 66, please also see the rest of the manuscript too, e.g. line 87-94 and more.
- The results and methods are inter-mixed
- Line 99 shows a period of 1984-2020 but Line 17 says that the study period was 1985-2020.
- How about Sentinel image band combinations ? after equation 1 and 2
- The methods are not clear
- Section 2.3.4: the authors need to state to develop a framework whereby the required methods are clearly mentioned, this section seems to be irrelevant here.
- Line 192: There is not need to advertise ENVI
- Line 202->" There are results inserted into the Methods section
- Table 2 is very basic and does not need to be highlighted with such details.
- Table 3 can well be presented in graphical format
- Section 3.2: this wasn't found in the results section, and the content under this section belongs to the methods section
- The presentation of the entire paper needs to be revised, the current format is very confusing and the authors failed to highlight the innovative part of this research. The way it is presented is very superficial and there are serious flaws which need to be corrected.
Round 2
Reviewer 1 Report
Dear authors,
many thanks for revising this manuscript. In my opinion, you have massively increased the quality of the manuscript.
Thank you very much for considering all of my comments.
I suggest publishing your manuscript without any further changes as it is.
Reviewer 2 Report
The authors addressed my concerns and I have no additional concerns.
Reviewer 3 Report
- One of the key areas the authors want to analyze is the drought as mentioned in the topic but this has largely been ignored in the subsequent text in the abstract, intro and methods, and even results
- Please depict the drought in your abstract and the rest of the paper
- The title shows that the authors intend to analyze the lake's physical properties but then only the lake's spatial extent has been focused on; therefore, modifications are suggested in the title and contents
- The authors need to state the indicators for drought, it is missing in the current version
- The methods and results have been intermixed.
- Section 2.3 comes under section 2.2..
- The authors have confined the "data" to only "rainfall" but they used Landsat and Sentinel products too which need to be depicted as the rainfall has been projected.
- The different sections of the paper are not synchronized, it is difficult to follow the flow.
- Figure 4, do you really think it is a method or result?
- Harmonize the text,vunits in vertical axes e.g. figure 5 and 7